# Using Hybrid Artificial Intelligence and Machine Learning Technologies for Sustainability in Going-Concern Prediction

**Der-Jang Chi and Zong-De Shen ***

Department of Accounting, Chinese Culture University, No. 55, Hwa-Kang Road, Yang-Ming-Shan, Taipei City 11114, Taiwan; qdz@ulive.pccu.edu.tw
*   Correspondence: szd2@ulive.pccu.edu.tw

**Abstract:** The going-concern opinions of certified public accountants (CPAs) and auditors are very critical, and due to misjudgments, the failure to discover the possibility of bankruptcy can cause great losses to financial statement users and corporate stakeholders. Traditional statistical models have disadvantages in giving going-concern opinions and are likely to cause misjudgments, which can have significant adverse effects on the sustainable survival and development of enterprises and investors' judgments. In order to embrace the era of big data, artificial intelligence (AI) and machine learning technologies have been used in recent studies to judge going concern doubts and reduce judgment errors. The Big Four accounting firms (Deloitte, KPMG, PwC, and EY) are paying greater attention to auditing via big data and artificial intelligence (AI). Thus, this study integrates AI and machine learning technologies: in the first stage, important variables are selected by two decision tree algorithms, classification and regression trees (CART), and a chi-squared automatic interaction detector (CHAID); in the second stage, classification models are respectively constructed by extreme gradient boosting (XGB), artificial neural network (ANN), support vector machine (SVM), and C5.0 for comparison, and then, financial and non-financial variables are adopted to construct effective going-concern opinion decision models (which are more accurate in prediction). The subjects of this study are listed companies and OTC (over-the-counter) companies in Taiwan with and without going-concern doubts from 2000 to 2019. According to the empirical results, among the eight models constructed in this study, the prediction accuracy of the CHAID–C5.0 model is the highest (95.65%), followed by the CART–C5.0 model (92.77%).

**Keywords:** going concern; artificial intelligence (AI); machine learning; classification and regression trees (CART); chi-squared automatic interaction detector (CHAID); extreme gradient boosting (XGB); artificial neural network (ANN); support vector machine (SVM); C5.0

## 1. Introduction

In recent years, in the competitive market environment, many enterprises have been bankrupt, causing serious losses to financial statement users and public investors, and thus the going-concern doubts of enterprises have received increased attention. If certified public accountants (CPAs) or auditors fail to give audit opinions on going-concern doubts before corporate bankruptcy, they will cause significant damages to themselves or their firms [1]. The going-concern assumption, which is one of the four basic accounting assumptions, means that enterprises will continue to operate in their current size and status for the foreseeable future; that is, enterprises are able to continue to operate for at least 12 months after the balance sheet date without being dissolved and liquidated. In the face of mass data and the age of artificial intelligence (AI), the Big Four accounting firms (Deloitte, KPMG, PwC, and EY) pay more and more attention to big data, AI, and machine-learning technologies.

The complex process to assess whether enterprises have going-concern doubts promotes the development of going-concern prediction models, which auditors can construct

with past financial and non-financial variables. In order to reduce the possibility of misjudgment and litigation risks, the model prediction results provide reference for auditors to decide whether to issue audit opinions of going-concern doubts in the future [2]. Auditing regulators worldwide require CPAs to add information to audit reports by discussing key audit matters (KAMs) [3]. The main reason for audit failure is the complex decision-making process of auditors' reasonable assessment of the going-concern assumption and is related to auditors' professional judgments [4–6]. However, enterprises and investors focus on whether the corporate governance mechanisms regulated by competent authorities can actually improve audit quality [1]. The viewpoint of different stakeholders about the prediction and evaluation of corporate financial distress, the way in which managers and auditors (or CPAs) influence and evaluate the communication of a company's financial distress, and accounting principles and regulations, will affect the evaluation of going concern [7]. In most situations, auditors have uncertainty about the going concern that led to the bankruptcy of companies, and the main underlying factors being reported losses, negative equity, and the business history. It is the responsibility of auditors and CPAs to understand whether each audit engagement is appropriate for the management's use of the going concern. But the auditor cannot be imputed to the situation in which the audited entity fails, despite the fact that the report has been issued without the additional going concern principle [8]. CPAs and their auditing teams may be affected by various factors, such as time pressure and remuneration, which may affect their going-concern opinions [9]. While all CPAs and auditors perform audits according to auditing standards and relevant laws, large international accounting firms (such as Deloitte, KPMG, PwC, and EY) have rigorous auditing systems and norms, which assist, guide, and supervise their CPAs and auditors to reduce the risks of audit failure; for example, the key audit matters (KAMs) of customers are examined by computer-assisted audit techniques, and walk-throughs, control tests, and substantive procedures are carried out during the audit process to improve the quality of audit reports and opinions. Even so, CPAs still issue incorrect audit reports and opinions from time to time. Therefore, it is necessary to use an effective going-concern prediction model to help CPAs make correct going-concern opinion decisions and reduce the risk of CPAs being punished or sued.

While most previous researchers have constructed going-concern decision models with traditional statistical methods, such methods have disadvantages and are likely to cause misjudgments. Thus, due to model complexity, some researchers have proposed constructing going-concern decision models by data mining and machine-learning technologies, among them, decision trees, artificial neural networks (ANNs), and support vector machines (SVMs), which are the most commonly used and have been applied in many different fields. Artificial neural networks have the advantages of parallel processing, high fault tolerance, and generalization ability. Decision trees can process missing values and avoid data overfitting through tree pruning, and because they do not require a large number of training, the generated models are easy to understand. Support vector machines are applied to determine the optimal separating hyperplanes that can classify input training data into two or more different classes through learning mechanisms.

In addition, previous researchers propose that variable selection in advance can decrease the data dimension, reduce noise interference, and improve classification accuracy. Therefore, in order to improve the classification accuracy of data mining methods, this study selects the influential variables in advance.

As traditional statistical models have disadvantages and high error rates in giving going-concern opinions, it is urgent to construct effective and accurate going-concern opinion decision models for auditors. Data mining and machine-learning technologies have been used in some recent studies to judge going-concern doubts and reduce judgment errors; however, the research literature remains inadequate. The purpose of this study is to propose suitable methods to construct going-concern opinion decision models in order to detect the signs of bankruptcy in advance and reduce losses to investors and auditors. This study integrates AI and machine-learning technologies, including the selection of

important variables by two decision trees: the chi-squared automatic interaction detector (CHAID) and classification and regression trees (CART); then, the classification models are respectively constructed by machine-learning technologies, such as extreme gradient boosting (XGB), artificial neural network (ANN), support vector machine (SVM), and C5.0, and the going-concern prediction accuracy of the models are compared to identify the optimal going-concern decision model. The highlights of this study are using hybrid artificial intelligence and several powerful and efficient machine-learning algorithms to construct going-concern prediction models.

The structure of this study is described in order as follows: Section 1. Introduction, Section 2. Literature Review, Section 3. Materials and Methods, Section 4. Results, Section 5. Discussion, and Section 6. Conclusions.

## 2. Literature Review

### 2.1. Definition of Going Concern

The going-concern assumption means that enterprises will continue to operate in their current size and status for the foreseeable future—that is, enterprises will be able to continue to operate for at least one year after the date of their balance sheet (US GAAP) or the date of the Statement of Financial Position (IFRSs) without being dissolved and liquidated [6]. As stated in SAS No. 57 of Taiwan [10], in the case of existing material uncertainties related to events or conditions that may cast significant doubts on audited enterprises' abilities to continue as going concerns, according to the risk assessment as required by IFRSs, CPAs shall issue reports in accordance with the statement on auditing standards. Based on the obtained audit evidence, they must conclude whether the management's use of the going-concern basis of accounting is appropriate and whether material uncertainties exist related to events or conditions that may cast significant doubts on audited enterprises' abilities to continue as going concerns [10]. CPAs will issue audit reports based on the aforementioned financial statements and risks: (1) unqualified opinion; (2) qualified opinion; (3) disclaimer opinion; and (4) adverse opinion [11]. If a company fails to fully disclose going-concern-related matters in notes to their financial statements or if CPAs determine that these matters are not fully disclosed, then CPAs will issue a qualified opinion or an adverse opinion. In Taiwan, CPAs' audit reports and opinions have the following effects on the trading of stocks: (1) unqualified opinion (normal trading); (2) qualified opinion (trading method changing); (3) disclaimer of opinion (suspension of trading); and (4) adverse opinion (suspension of trading).

According to SAS No. 61 of Taiwan [11], the preparation of financial statements is usually based on the going-concern assumption, and auditors shall assess the reasonability of the going-concern assumption in accordance with the statement. After the assessment on the reasonability of the going-concern assumption, CPAs may issue audit reports with unqualified opinions if doubts can be eliminated and shall issue audit reports with qualified or adverse opinions if they consider that future measures are reasonable but must be disclosed in the financial statements. If the doubts of the going-concern assumptions of audited enterprises cannot be eliminated, but are appropriately disclosed in their financial statements, then CPAs shall issue audit reports with qualified or adverse opinions according to the effects. If CPAs determine that the going-concern assumptions on which the financial statements of audited enterprises are prepared do not conform to the actual situations and have significant effects, then they shall issue audit reports with adverse opinions [11]. Therefore, in the case of failure to eliminate CPAs' doubts or inconsistency with the actual situations, explanations shall be added to the audit reports of audited enterprises, and such opinions are called going-concern opinion audit reports. In recent years, Taiwan has been revising its laws and rules on CPAs' reports for the following purposes: (1) Provide transparency of an audit on financial information; (2) Improve users' awareness of key audit matters in the report; and (3) Increase users' understanding of the management's key judgment matters in the report. Taiwan has a special system, known as the double

signature system, which means that the report of a listed (OTC) company must be signed (audited) by two CPAs.

In order to increase the responsibilities of regulatory authorities and CPAs and to prevent frauds, the United States Congress passed the Sarbanes-Oxley Act in 2002. Based on the principles and norms of the Sarbanes-Oxley Act, CPAs and auditors are required to collect sufficient and appropriate evidence during auditing to determine whether audit reports with going-concern doubts should be issued to audited enterprises. CPAs and auditors shall avoid direct or indirect contributions to fraud in the financial statements of corporate management due to willful misconduct or gross negligence. The main purpose of auditors in financial-statement auditing is not to assess enterprises' abilities to continue as going concerns, but rather to confirm whether there is any material misrepresentation in their financial statements. However, if going-concern doubt reports are not issued before a bankruptcy crisis, then it is often regarded as an audit failure by financial statement users and public investors [4–6,12]. In order to enhance the supervision effectiveness of boards based on their independence and professionalism and regain investors' confidence in the capital market, Taiwan has actively strengthened its corporate governance mechanisms since 2002.

### 2.2. Statistical Methods of Going-Concern Decisions and Related Works

In the past, most researchers study going-concern decisions by traditional statistical methods, such as regression analysis, factor analysis, cluster analysis, and discrimination analysis; however, these methods have disadvantages and are likely to cause misjudgments. Thus, data mining and machine learning technologies have been used in some recent studies to judge going-concern doubts and reduce judgment errors. The studies involve research on going-concern decisions by artificial neural networks (ANNs) [4–6,12–14], by decision trees [4,6,12–16], by Bayesian belief networks (BBNs) [12,14], and by support vector machines (SVMs) [15,17–19]. In recent years, there have been some important studies on going-concern decisions by machine learning or deep learning. Barboza et al. [20] utilize machine-learning methods (support vector machines, bagging, boosting, neural networks, and random forest) to predict bankruptcy and compare their performance with results from discriminant analysis and logistic regression. Their results show machine-learning models, on average, approximately 10% more accuracy in relation to traditional models. Comparing the best models, the random forest model (machine-learning technique) led to 87% accuracy, whereas logistic regression and linear discriminant analysis only led to 69% and 50% accuracy. Goo et al. [6] use three machine-learning technologies, such as the least absolute shrinkage and selection operator (LASSO) and neural network (NN), classification and regression tree (CART), and support vector machine (SVM), to construct predictions for going-concern models, among which the prediction accuracy of the LASSO–SVM model is the highest (89.79%). Chen [1] selects important variables by stepwise regression (SR), support vector machine (SVM), and artificial neural network (ANN) in the first stage, and uses the classification and regression tree (CART), chi-square automatic interaction detector (CHAID), C5.0, and quick unbiased efficient statistical tree (QUEST), respectively, in the second stage to construct going-concern prediction models, among which the prediction accuracy of the SR–CHAID model is the highest (89.03%). Pawełek [21] adopts the extreme gradient boosting method to predict company bankruptcy. The results show that the use of quantiles for the removal of the outliers by going-concern companies from the training set improves the accuracy of the extreme gradient-boosting methods in detecting bankrupt companies. Chen and Shen [22] use multiple machine-learning techniques, firstly selecting important variables with stepwise regression (SR) and the least absolute shrinkage and selection operator (LASSO), and then using classification and regression tree (CART) and random forests (RF) to build predictive models. Financial and non-financial variables are used in their study. The results show that the highest prediction accuracy rate of financial distress is up to 89.74% with the LASSO–CART model. Jan [16] uses the classification and regression tree (CART), deep neural network (DNN), and recurrent neural network (RNN)

to construct going-concern prediction models, among which, the prediction accuracy of the CART–RNN model is the highest (95.28%). Chi and Chu [23] use long short-term memory (LSTM) and gated recurrent unit (GRU) to construct going-concern prediction models, among which, the prediction accuracy of the LSTM model is the highest (96.15%). In these important studies of going-concern decisions by machine learning or deep learning, the accuracy of the constructed going-concern prediction models is higher than 80%, which greatly inspires this study.

## 3. Materials and Methods

As two technologies with very strong selection and classification abilities in decision tree algorithms, CART and CHAID are suitable for selecting important variables. XGB, ANN, SVM, and C5.0 are four machine-learning algorithms with good learning and prediction abilities. Hence, this study first selects important variables by CART and CHAID, and then constructs going-concern prediction models by XGB, ANN, SVM, and C5.0.

### 3.1. Classification and Regression Tree (CART)

The classification and regression tree (CART) is a data mining and prediction algorithm developed by Breiman et al. [24]. As a decision-tree technology based on binary segmentation, it is used for continuous or classified non-parameter data, where the segmentation conditions are determined according to the number of classes, data attributes, and Gini rules (minimum Gini). The data are segmented into two subsets, and then the procedure is repeated to identify the next segmentation condition in each subset. Therefore, if data set $S$ has $N$ pieces of data and can be classified into $i$ classes, and the sample size of each class is $n_i$, Gini ($G$) can be calculated, as shown in Equation (1). Moreover, if the data set is classified into two subsets ($S_1$, $S_2$) by a given variable $x$, Gini ($G$) can be calculated by Equation (2). The higher the Gini, the less pure the information.

$$G(S) = 1 - \sum_{i=1}^{i} \left( \frac{n_i}{N} \right)^2 \tag{1}$$

$$G(S,x) = \left( \frac{S_1}{S} \right) \times G(S_1) + \left( \frac{S_2}{S} \right) \times G(S_2) \tag{2}$$

### 3.2. Chi-Squared Automatic Interaction Detection (CHAID)

As its name implies, chi-squared automatic interaction detection (CHAID) is used to calculate the $p$-value of branch-leaf splitting nodes in a decision tree by chi-square testing, in order to decide whether to continue segmentation. The closer the relationship between characteristic variables and class variables, the smaller the $p$-value will be, and such characteristic variables will be selected as those with optimal classification. CHAID can prevent data from being overused and stop segmentation by the decision tree; that is, CHAID finishes pruning before the models are completed. Chi-square values are used to measure the bias between actual values and theoretical values of samples; the larger the bias, the larger the chi-square value; the smaller the bias, the closer the chi-square value will approach.

### 3.3. Extreme Gradient Boosting (XGB)

XGB, as proposed by Chen and Guestrin [25], is an extension of the gradient-boosting decision tree proposed by Breiman. In the analysis process, each calculation is intended to reduce the residual of the previous calculation, in other words, each new model reduces the residual of the previous model. The advantages of XGB include [21,25,26]: (1) it can do both classification and regression continuous value prediction, and the effect is usually quite good; (2) the loss function is expanded by Taylor expansion, and the first-order and second-order derivatives are used at the same time, which can speed up the optimization; (3) a sub-feature-extraction is introduced (does not need to use all features for training),

like that of random forest, it can avoid over-fitting and reduce computation time; (4) the use of a local approximation algorithm to optimize split nodes; (5) provides GPU parallelization. The XGB-based prediction model summarizes the results of each tree, as shown in Equation (3). In $F = \{f(x) = wq(x)\}$, q represents the leaf nodes of each tree structure, and K represents the number of constructed trees.

$$\hat{y}_i = \sum_{k=1}^{K} f_k(x_i), f_k \in F \tag{3}$$

The objective function is composed of a loss function and a regularizer, as shown in Equation (4). The regularizer is a penalty term, as shown in Equation (5), which has the main function of solving the problem of overfitting. *T* is the number of leaf nodes, *Wj* is the score of node *J*, and $\gamma$ and $\lambda$ are penalty coefficients.

$$L(\Phi) = \sum_{k=1}^{K} l(\hat{y}_l, y_i) + \sum_{k=1}^{K} \Omega(f_k(x_i)) \tag{4}$$

$$\Omega(f_k(x_i)) = \gamma T + \frac{1}{2}\lambda\sum_{J=1}^{T} W_j^2 \tag{5}$$

The loss function used for classification is shown in Equation (6):

$$Logistic : -[(y_i \log \hat{p}_l) + (1 - y_i)(\log(1 - \hat{p}_l))] \tag{6}$$

In terms of the feature selection rules of XGB, as shown in Equation (7), larger scores after segmentation are better; therefore, those with large Gain values were selected for segmentation.

$$Gain = \frac{1}{2}\left[\frac{G_L^2}{H_L + \lambda} + \frac{G_R^2}{H_R + \lambda} - \frac{(G_L + G_R)^2}{H_L + H_R + \lambda}\right] - \gamma \tag{7}$$

### 3.4. Artificial Neural Network (ANN)

The theory of artificial neural networks originated in the 1950s. Proposed by Hopfield [27], an artificial neural network (ANN) is a parallel computation model similar to the human neural structure and an information processing technology inspired by the study of the brain and nervous system. It is commonly known as a parallel distributed processing model or a link model. ANN can build a system model (the relationship between input and output) by means of a set of examples, namely, the data composed of the input and output of the system [28], and this system model can be used for estimation, prediction, decision, and diagnosis. As common statistical techniques for regression analysis can be used, artificial neural networks can be considered as a special form of statistical techniques. In terms of the operation, the predicted value $\hat{y}$ is estimated based on the transmission by neurons in the hidden layer. If the independent variables are $X_1, X_2, X_3, \ldots\ldots, X_n$, their weights are $\omega_1, \omega_2, \omega_3, \ldots\ldots, \omega_n$, the startup function is $\sigma$, and the deviation value is $b$. The operation of an artificial neural network is shown in Equation (8).

$$\hat{y} = \sigma(X_1\omega_1 + X_2\omega_2 + X_3\omega_3 + X_n\omega_n + b) \tag{8}$$

### 3.5. Support Vector Machine (SVM)

The support vector machine (SVM) is a method proposed by Vapnik [29] and is derived from the statistical learning theory and structural risk minimization (SRM), which have been developed from simple vector classifiers to hyperplane classifiers and can be classified into linear- and nonlinear-support vector machines according to the different problems processed.

The basic operational concept of a support vector machine is that the input vector is mapped into a high-dimensional feature space by a linear or non-linear kernel function to determine the optimal hyperplane in the feature space to distinguish all classes. Therefore, the problems that cannot be solved linearly in low dimensions can be classified in high

dimensions, and the feature space composed of high dimensions can also be infinite. Through appropriate kernel functions, nonlinear images can help decision functions solve the problems in the new feature space, and this feature enabled Vapnik to apply the minimum structural risk to nonlinear problems while still using the optimal technique. The decision functions determined by support vector machines are composed of special vectors, which are known as support vectors [30,31], and are selected from training data; therefore, this method is known as a support vector machine. SVM classifies *I* training data into two categories with a hyperplane, which can be defined as Equation (9):

$$f_H(X) = w \cdot x + b \tag{9}$$

The optimal w is marked as $\overline{w}$, and the solution is shown in Equation (10):

$$\overline{w} = \sum_{i=1}^{l} \overline{\alpha}_i y_i x_i \tag{10}$$

The linear kernel is used in this study to construct the SVM classification model, and the kernel function equation is shown in Equation (11):

$$K(x, x') = x^T x' \tag{11}$$

### 3.6. C5.0

C5.0 is an improvement on the Iterative Dichotomiser 3 (ID3), as proposed by Quinlan [32] in 1986, as ID3 is unable to process continuous data. C5.0 first treats data as the same group, then selects branch attributes to calculate information gains, in order to identify the best attributes. C5.0 has many advantages, including (1) being steady in the face of missing data and input field; (2) not requiring high training frequency; (3) the model is easy to understand; (4) allowing multiple segmentations with more than two subgroups; and (5) providing a powerful enhancement technology to improve classification accuracy.

### 3.7. Models' Performance Evaluation Methods

In order to compare the prediction performance among the models, this study uses the accuracy rate as Equation (12) and the GC sample prediction error rate as Equation (13) to measure the prediction performance of all models.

$$Accuracy = \frac{number\ of\ correct\ predictions}{total\ number\ of\ predictions} \tag{12}$$

$$GC\ error\ rate \frac{false\ positive\ predictions}{total\ number\ of\ correct\ predictions} \tag{13}$$

The confusion matrix [16,23] is also used in this study, the indicators of confusion matrix are accuracy, precision as Equation (14), recall (sensitivity) as Equation (15), and F1-score as Equation (16).

$$precision = ture\ postive/(ture\ postive + false\ postive) \tag{14}$$

$$recall = ture\ positive/(ture\ postive + false\ negative) \tag{15}$$

$$F1 - score = 2 \times precision \times recall/(precision + recall) \tag{16}$$

### 3.8. Sampling and Variable Selection
3.8.1. Data Sources

The sample period in this study is 20 years from 2000 to 2019, listed and OTC (over-the-counter) companies issued with audit opinions of going-concern doubts are taken as the sample subjects, and the companies issued with audit opinions of going-concern doubts

in the first year of the study period are selected as the samples. The variables are sourced from the database of the Taiwan Economic Journal (TEJ).

In order to eliminate the influence of many external environmental factors, such as time, industry, and company size, pairing is used to control the factors in this study; hence, the concept of paired samples, as proposed by [1,6,33], is adopted. Companies without going-concern doubts in the same year and same industry, and whose total assets are similar to those of companies issued with audit opinions of going-concern doubts in the previous year, are considered as paired samples. Hence, in this study, one sample company with going-concern doubts is paired with three sample companies without going-concern doubts (with doubts: without doubts = 1:3). There are a total of 536 companies, and the industry distribution is shown in Table 1.

**Table 1.** Industry distribution of samples.

| Industry | with Going-Concern Doubts (GCD) | without Going-Concern Doubts (Non-GCD) | Total |
|---|---|---|---|
| Cement | 1 | 3 | 4 |
| Food | 1 | 3 | 4 |
| Textile fiber | 6 | 18 | 24 |
| Electromechanics | 5 | 15 | 20 |
| Electric appliances and cables | 2 | 6 | 8 |
| Chemical | 1 | 3 | 4 |
| Biotechnology and medical treatment | 6 | 18 | 24 |
| Steel | 8 | 24 | 32 |
| Semiconductor | 15 | 45 | 60 |
| Computers and peripherals | 6 | 18 | 24 |
| Opto-electronics | 24 | 72 | 96 |
| Communication network | 1 | 3 | 4 |
| Electronic components | 12 | 36 | 48 |
| Electronic circuit | 1 | 3 | 4 |
| Information services | 3 | 9 | 12 |
| Other electronics | 4 | 12 | 16 |
| Building materials and construction | 12 | 36 | 48 |
| Shipping | 1 | 3 | 4 |
| Tourism | 4 | 12 | 16 |
| Trade | 3 | 9 | 12 |
| Cultural and creative | 4 | 12 | 16 |
| Agricultural science and technology | 1 | 3 | 4 |
| Oil, electricity, and gas | 1 | 3 | 4 |
| Others | 12 | 36 | 48 |
| Total | 134 | 402 | 536 |

### 3.8.2. Variable Characteristics and Definitions

This study uses a total of 26 input variables, of which GC (going concern) is used as the target variable. Any company issued a going-concern doubt is 1, and any company not issued the audit opinion of going-concern doubt is 0. $X1$-$X25$ are input variables, except for $X25$, which is a categorical variable, and the rest ($X1$-$X24$) are continuous variables.

1.  Dependent variables

Dependent variables are classified according to whether the companies are issued with going-concern doubts. Any company issued with a going-concern doubt is 1, and any company not issued with the audit opinion of going-concern doubt is 0.

2. Independent variables

A total of 25 variables commonly used to measure going concern are selected in this study, including 19 financial variables (X1-X19) and 6 non-financial variables (X20-X25), which are summarized in Table 2.

**Table 2.** Research variables.

| Code | Variable Name | Variable Definition or Calculation Equation | Sources |
|------|---------------|---------------------------------------------|---------|
| X1 | Natural log of total assets | Ln (total assets) | Chen [1]; Chen and Lee [5]; Goo et al. [6]; Jan [33]; Chen and Jhuang [34] |
| X2 | Debt ratio | Total liabilities/total assets | Chen [1]; Chen and Lee [5]; Goo et al. [6]; Jan [16]; Chen and Shen [22]; Jan [33]; Chen and Jhuang [34]; Jan [35]; Jan [36] |
| X3 | Quick ratio | Quick assets/current liabilities | Chen [1]; Jan [16]; Pawełek [21]; Chen and Shen [22]; Jan [33]; Chen and Jhuang [34]; Jan [35]; Jan [36] |
| X4 | Current ratio | Current assets/current liabilities | Chen [1]; Chen and Lee [5]; Goo et al. [6]; Jan [16]; Chen and Shen [22]; Chi and Chu [23]; Jan [33]; Jan [35]; Jan [36] |
| X5 | Total liabilities/ stockholders' equity | Total liabilities/stockholders' equity | Chen [1]; Jan [16]; Chen and Shen [22]; Jan [36] |
| X6 | Current liabilities/total liabilities | Current liabilities/total liabilities | Chen and Lee [5]; Jan [16]; Chi and Chu [23]; Jan [35] |
| X7 | Current assets/total assets | Current assets/total assets | Chen [1]; Chen and Lee [5]; Chi and Chu [23]; Jan [35] |
| X8 | Fixed assets/total assets | Fixed assets/total assets | Chi and Chu [23] |
| X9 | Fixed assets/long-term liabilities | Fixed assets/long-term liabilities | Chi and Chu [23]; Jan [35] |
| X10 | Stockholders' equity/ fixed assets | Stockholders' equity/fixed assets | Pawełek [21]; Chi and Chu [23] |
| X11 | Proportion of long-term funds in fixed assets | (Stockholders' equity + long-term liabilities)/fixed assets | Jan [16]; Chen and Shen [22]; Jan [35] |
| X12 | Return on total assets | Net profit after tax +interest*(1-tax rate)/ average total assets | Chen [1]; Chen and Shen [22]; Jan [33]; Jan [35]; Jan [36] |
| X13 | Return on stockholders' equity | Net profit after tax/average stockholders' equity | Chen [1]; Chen and Shen [22]; Jan [33]; Jan [35]; Jan [36] |
| X14 | Debt dependence | Long-term and short-term loans/ stockholders' equity | Jan [35]; Jan [36] |
| X15 | Total assets turnover | Net operating income/average total assets | Chen [1]; Jan [16]; Chen and Jhuang [34] |
| X16 | Accounts receivable turnover | Net operating income/ average accounts receivable and notes | Goo et al. [6]; Chen and Shen [22]; Chen and Jhuang [34] |
| X17 | Inventory turnover | Cost of goods sold/average inventory | Goo et al. [6]; Chen and Shen [22] |
| X18 | Fixed assets turnover | Net operating income/average fixed assets | Chen and Jhuang [34] |
| X19 | Earnings per share | Market value per share/earnings per share | Jan [16]; Chen and Jhuang [34]; Jan [35] |
| X20 | Shareholding ratio of directors and supervisors | Total number of stocks held by directors and supervisors/capital stock | Chen and Lee [5]; Chen and Shen [22]; Chi and Chu [23] |
| X21 | Shareholding ratio of managers | Number of stocks held by managers/capital stock | Jan [36] |
| X22 | Stockholding ratio of major shareholders | Number of stocks held by major shareholders/ capital stock | Jan [16]; Chen and Shen [22]; Chi and Chu [23]; Jan [35]; Jan [36] |
| X23 | Pledge ratio of directors and supervisors | Number of stocks pledged by directors and supervisors/number of stocks held by directors and supervisors | Chen and Lee [5]; Chen and Shen [22]; Chi and Chu [23]; Jan [35]; Jan [36] |
| X24 | Proportion of directors serving as managers | Number of directors serving as managers/total number of directors | Chen [1]; Chen and Shen [22]; Jan [33] |
| X25 | Audited by BIG4 or not | 1 for companies audited by BIG4, otherwise 0 | Chen [1]; Chen and Lee [5]; Goo et al. [6]; Jan [16]; Chen and Shen [22]; Jan [33]; Jan [35] |

### 3.9. Research Design and Process

First, through literature review and practice, this study selects 25 variables that may affect the audit opinions of going-concern doubts, and then the influential variables are selected by CART and CHAID algorithms in decision trees. Then, after the going-concern doubt prediction models are constructed by XGB, ANN, SVM, and C5.0, the classification accuracy, as well as the proportion of samples with going-concern doubts (GCD) that are mispredicted as those without going-concern doubts (Non-GCD), are added to the total samples. The research design and process are shown in Figure 1.

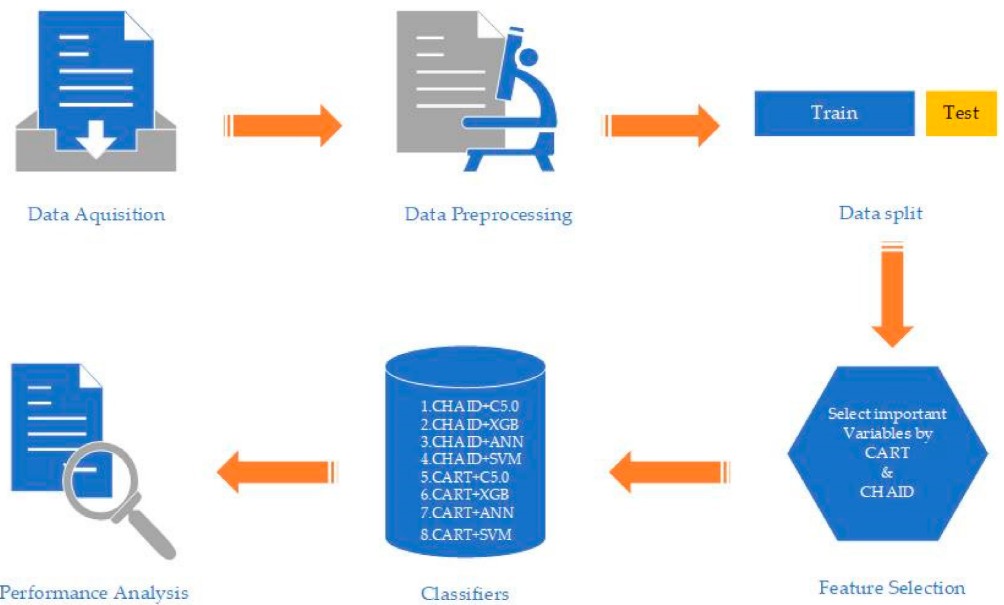

**Figure 1.** Research design and process.

## 4. Results

### 4.1. Variable Selection Results

This study uses the decision trees, CART and CHAID, as the tools to select important variables in the first stage, reduce noise, and improve model accuracy. Then, seven variables are selected by CART in the order of their importance (numbers in brackets are the significance values of the variables): X2 debt ratio (0.26), X13 return on stockholders' equity (0.19), X12 return on total assets (0.12), X14 debt dependence (0.09), X19 earnings per share (0.09), X5 total liabilities/stockholders' equity (0.07), and X23 pledge ratio of directors and supervisors (0.03), as shown in Table 3. Then, eight variables are selected by CHAID in the order of their importance (numbers in brackets are the significance values of the variables): X5 total liabilities/ stockholders' equity (0.54), X19 earnings per share (0.19), X12 return on total assets(0.16), X25 audited by BIG4 or not(0.03), X16 accounts receivable turnover(0.03), X13 return on stockholders' equity(0.02), X18 fixed assets turnover(0.02), and X1 natural log of total assets(0.01), as shown in Table 4.

**Table 3.** Variables selected by CART.

| Variables | Importance |
|---|---|
| X2 Debt ratio | 0.26 |
| X13 Return on stockholders' equity | 0.19 |
| X12 Return on total assets | 0.12 |
| X14 Debt dependence | 0.09 |
| X19 Earnings per share | 0.09 |
| X5 Total liabilities/stockholders' equity | 0.07 |
| X23 Pledge ratio of directors and supervisors | 0.03 |

**Table 4.** Variables selected by CHAID.

| Variables | Importance |
|---|---|
| X5 Total liabilities/stockholders' equity | 0.54 |
| X19 Earnings per share | 0.19 |
| X12 Return on total assets | 0.16 |
| X25 Audited by BIG4 or not | 0.03 |
| X16 Accounts receivable turnover | 0.03 |
| X13 Return on stockholders' equity | 0.02 |
| X18 Fixed assets turnover | 0.02 |
| X1 Natural log of total assets | 0.01 |

*4.2. Model Construction Results*

As mentioned above, this study samples 536 listed and OTC companies in Taiwan over 20 years (from 2000 to 2019), identified 25 research variables (19 financial variables and 6 non-financial variables) and collects sufficient data. All of the data is normalized and standardized between 0 and 1 [16,23,35,36]. This study randomly assigns 70% of the data types selected by CART and CHAID as the training group, in order to construct going-concern doubt prediction models by XGB, ANN, SVM, and C5.0, while the remaining 30% are used as the test group to validate the classification abilities (prediction abilities) of the models. According to Chen [1] and Jan [33], models constructed in this manner have validity.

In this study, the parameters of the classification models are set as follows. The SVM adopts a linear kernel, the penalty function C is 1, and the epsilon is 0.1; for the ANN, the hidden layer is 2 layers, and the first layer is 7 neurons, the second layer is 32 neurons, and the number of training is 2000 times; for the XGB, the learning rate is 0.2, the gamma is 0.1, and there is no limit to the development of the tree structure, so it is 0, the proportion of random-sampling columns used to control each tree structure is 0.25, and the min child weight part is 1; and for the C5.0, the number of boosting is 10, the pruning severity is 80, and the minimum records per child branch is 2.

4.2.1. CART Models

The variables selected by CART are used to construct the XGB, ANN, SVM, and C5.0 models. Regarding the test group, as shown in Table 5, the prediction accuracy of the CART–C5.0 model is the highest (92.77%), followed by the CART–XGB model (89.44%), the CART–ANN model (88.82%), and the CART–SVM model (87.86%).

**Table 5.** The performance of CART models.

| Models | Accuracy | | Proportion of GCD Samples Mispredicted as Non-GCD Samples in the Total Samples in the Test Group |
|---|---|---|---|
| | Training Group | Test Group | |
| CART–XGB | 93.60% | 89.44% | 5.59% |
| CART–ANN | 92.00% | 88.82% | 8.70% |
| CART–SVM | 89.13% | 87.86% | 4.05% |
| CART–C5.0 | 94.02% | 92.77% | 3.01% |

The economic damage caused on information users by misjudging samples with going-concern doubts (GCD) as those without going-concern doubts (Non-GCD) is far greater than that caused by misjudging samples without going-concern doubts as those with going-concern doubts, thus, this study analyzes and compares the proportion of GCD samples misjudged as Non-GCD samples. The error rate of the CART–C5.0 model is the lowest in this respect (3.01%), followed by the CART–SVM model (4.05%), the CART–XGB model (5.59%), and the CART–ANN model (8.70%).

### 4.2.2. CHAID Models

The variables selected by CHAID are used to construct the XGB, ANN, SVM, and C5.0 models. In the test group, as shown in Table 6, the prediction accuracy of the CHAID–C5.0 model is the highest (95.65%), followed by the CHAID–XGB model (88.82%), the CHAID–ANN model (88.67%), and the CHAID–SVM model (83.54%). By applying the same method described above, according to the comparison of the number of GCD samples mispredicted as Non-GCD samples, the error rate of the CHAID–C5.0 model is the lowest (1.24%), followed by the CHAID–ANN model (2.00%), CHAID–SVM model (6.10%), and CHAID–XGB model (7.45%).

**Table 6.** The performance of CHAID models.

| Models | Accuracy | | Proportion of GCD Samples Mispredicted as Non-GCD Samples in the Total Samples in the Test Group |
|---|---|---|---|
| | Training Group | Test Group | |
| CHAID–XGB | 92.53% | 88.82% | 7.45% |
| CHAID–ANN | 89.28% | 88.67% | 2.00% |
| CHAID–SVM | 85.05% | 83.54% | 6.10% |
| CHAID–C5.0 | 96.25% | 95.65% | 1.24% |

According to Table 5, further comparison of the test group of the CART–C5.0 model shows that the proportion of GCD samples mispredicted as Non-GCD samples in the total samples is the lowest (3.01%). According to Table 6, in the CHAID–C5.0 model test group, the proportion of GCD samples mispredicted as Non-GCD samples in the total samples is also the lowest (1.24%). In addition, regarding the prediction models constructed using C5.0 as a classifier, both the CHAID–C5.0 model (the accuracy of the test group is 95.65%), which is constructed to be used with CHAID for variable selection, and the CART–C5.0 model (the accuracy of the test group is 92.77%), which is constructed to be used with CART, performed well. In the test group, the proportions of GCD samples mispredicted as Non-GCD samples in the total samples are 1.24% and 3.01%, which are stable.

About the flowchart of the CHAID structure, the tree classification rule and structure generated, exampled by the CHAID–C5.0 model which has the highest prediction accuracy, as shown in Figure 2.

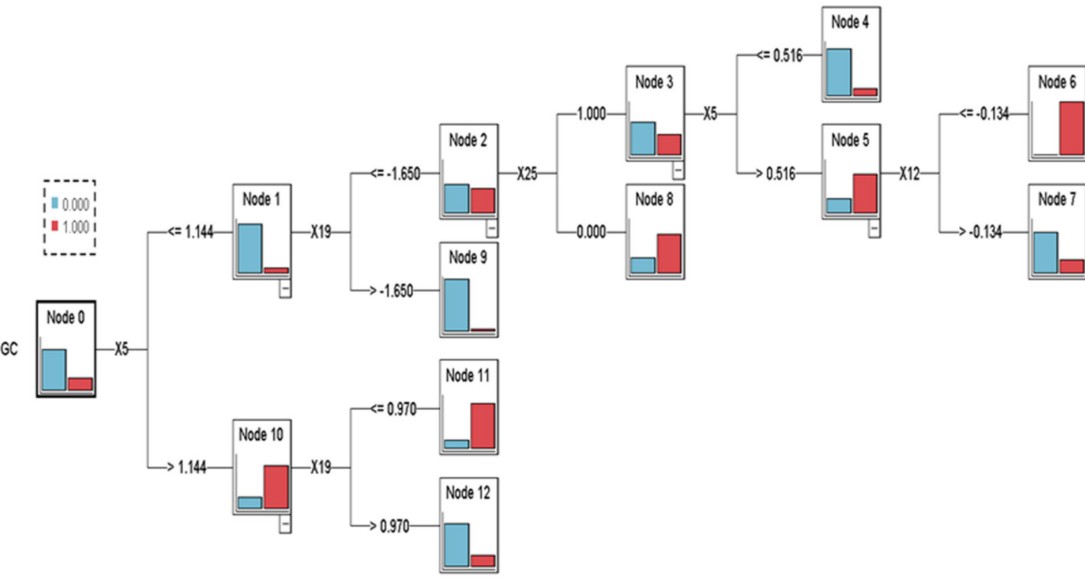

**Figure 2.** Decision path and structure of the CHAID–C5.0 model.

### 4.3. Additional Comparison of CHAID–C5.0 and CART–C5.0

Table 7 summarizes the comparison of model performances of CHAID–C5.0 and CART–C5.0. The confusion matrix indicators are accuracy, precision, recall (sensitivity), and F1-score. The CHAID–C5.0 model is better than the CART–C5.0 in all indicators. According to the most frequently used model performance indicator, the CHAID–C5.0 model reports an accuracy of 95.65% and the CART–C5.0 model at 92.77%. The CHAID–C5.0 model and the CART–C5.0 model also have low the proportions of GCD samples mispredicted as Non-GCD samples in the total samples, based on the results with the test group. This shows that the prediction error rate is very low, which can effectively reduce the risks and costs related to audit failures. The CHAID–C5.0 model and CART–C5.0 model for going-concern prediction constructed in this study are reliable and effective.

**Table 7.** The confusion matrix: comparison of model performances of CHAID–C5.0 and CART–C5.0.

| Indicators | CART–C5.0 | CHAID–C5.0 |
|---|---|---|
| Accuracy | 92.77% | 95.65% |
| Precision | 88.37% | 95.00% |
| recall | 84.44% | 88.37% |
| F1-score | 86.36% | 91.57% |

Finally, the ROC curve and AUC value are also adopted to compare the CHAID–C5.0 and CART–C5.0, as shown in Figure 3, Figure 4 and Table 8. The results also show that the CHAID–C5.0 model (AUC value: 0.957) is better than the CART–C5.0 model (AUC value: 0.912).

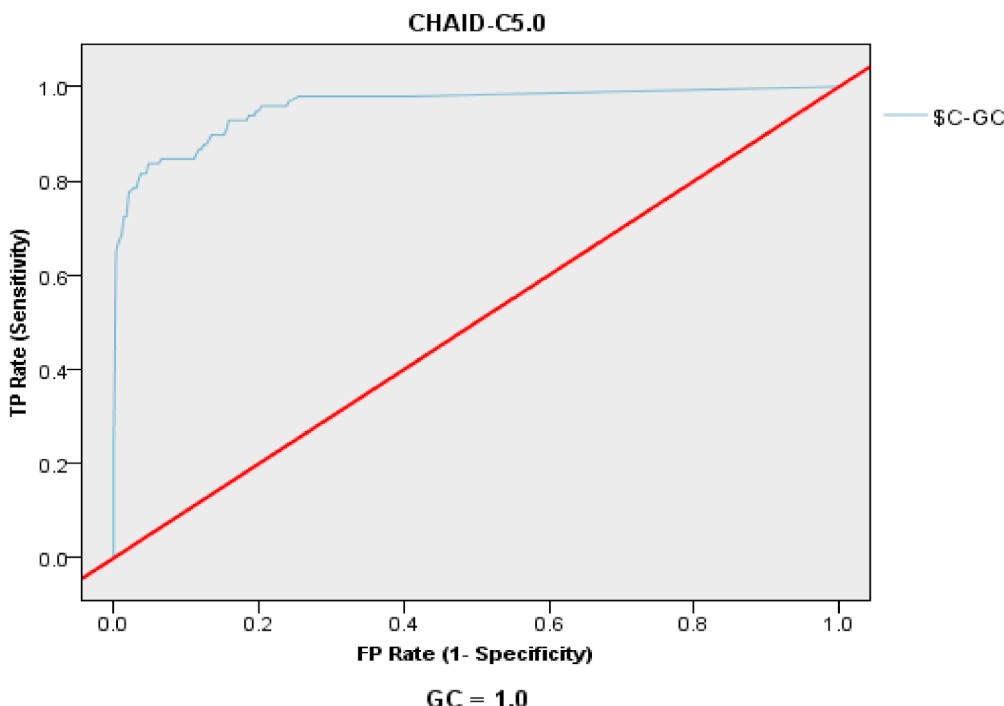

**Figure 3.** The ROC curve of CHAID–C5.0.

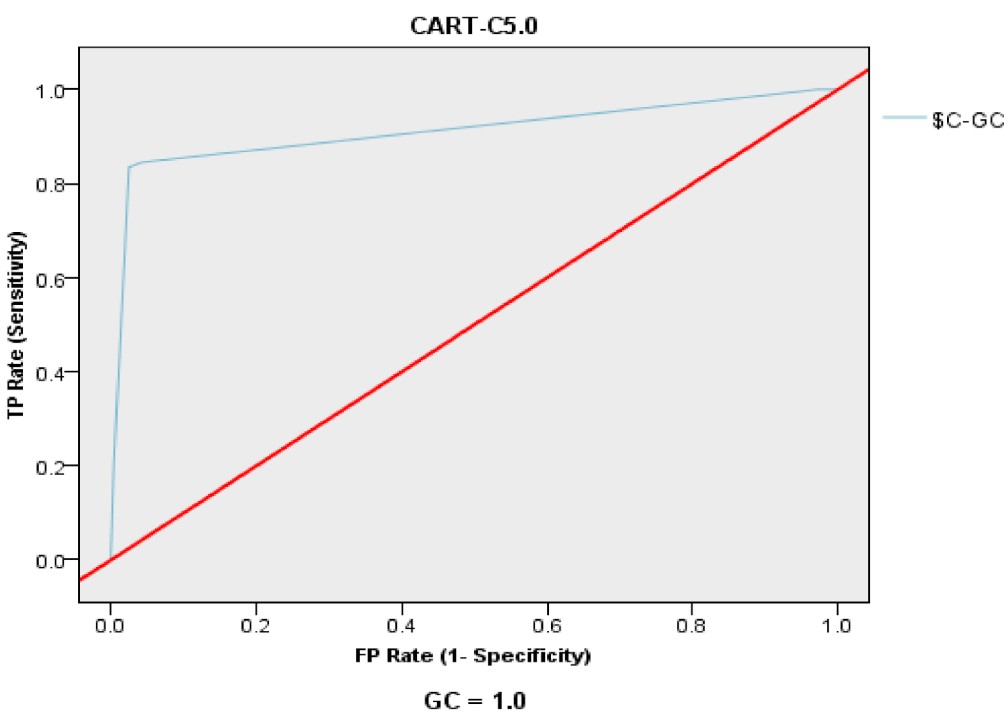

**Figure 4.** The ROC curve of CART–C5.0.

**Table 8.** The AUC value of CHAID–C5.0 and CART–C5.0.

| Models | CART–C5.0 | CHAID–C5.0 |
|---|---|---|
| AUC value | 0.912 | 0.957 |

## 5. Discussion

CART and CHAID are two technologies with very strong selection and classification abilities in decision tree algorithms. This study uses CART and CHAID as the tools to select important variables in the first stage, and then, four machine learning technologies with good learning and prediction abilities are applied, namely XGB, ANN, SVM, and C5.0, which are used to construct the going-concern prediction models. CART, CHAID, XGB, ANN, SVM, and C5.0 are widely used in machine learning technologies with strong classification abilities.

This study selects 19 financial variables and 6 non-financial variables that may affect the audit opinions of going-concern doubts through literature review and practice (as shown in Table 2). Based on the basic concept of statistics, modeling after selecting important variables from many variables will improve the prediction accuracy; hence, CART and CHAID are used in this study to select important variables. CART selects seven variables in the order of their importance, X2 debt ratio, X13 return on stockholders' equity, X12 return on total assets, X14 debt dependence, X19 earnings per share, X5 total liabilities/stockholders' equity, and X23 pledge ratio of directors and supervisors, as shown in Table 3. CHAID selects eight variables in the order of their importance, total liabilities/ stockholders' equity, X19 earnings per share, X12 return on total assets, X25 audited by BIG4 or not, X16 accounts receivable turnover, X13 return on stockholders' equity, X18 fixed assets turnover, and X1 natural log of total assets, as shown in Table 4. Then, 70% of the data are randomly drawn as the training group to construct the going-concern doubt prediction models of XGB, ANN, SVM, and C5.0, while the remaining 30% are used as the test group to construct the going-concern doubt prediction models by XGB, ANN, SVM, and C5.0.

The variables selected by CART are used to construct the CART–XGB, CART–ANN, CART–SVM, and CART–C5.0 models. As shown in Table 5, in the test group, the prediction accuracy of the CART–C5.0 model is the highest (92.77%), followed by the CART–XGB model (89.44), the CART–ANN model (88.82%), and the CART–SVM model (87.86%). In the test group, regarding the proportion of GCD samples mispredicted as Non-GCD samples in the total samples, the error rate of the CART–C5.0 model is the lowest (3.01%), followed by the CART–SVM model (4.05%), CART–XGB model (5.59%), and CART–ANN model (8.70%). According to the empirical results, the prediction accuracies of the CART–XGB, CART–C5.0, CART–ANN, and CART–SVM models are higher than 87%, and the proportion of GCD samples mispredicted as Non-GCD samples in the total samples in the test group is lower than 9%, which shows good results in terms of social science.

The variables selected by CHAID are used to construct the CHAID–XGB, CHAID–ANN, CHAID–SVM, and CHAID–C5.0 models. As shown in Table 6, in the test group the prediction accuracy of the CHAID–C5.0 model is the highest (95.65%), followed by the CHAID–XGB model (88.82%), the CHAID–ANN model (88.67%), and the CHAID–SVM model (83.54%), as shown in Table 6. In the test group, in terms of the proportion of GCD samples mispredicted as Non-GCD samples in the total samples, the error rate of the CHAID–C5.0 model is the lowest (1.24%), followed by the CHAID–ANN model (2.00%), CHAID–SVM model (6.10%), and CHAID–XGB model (7.45%). According to the empirical results, the prediction accuracy is higher than 83% for the CHAID–C5.0, CHAID–XGB, CHAID–ANN, and CHAID–SVM models, and the proportion of GCD samples mispredicted as Non-GCD samples in the total samples in the test groups is lower than 8%, which show good results. Among the eight models constructed in this study, the prediction accuracy of the CHAID–C5.0 model is the highest (95.65%). In order to determine which models are better, as constructed based on variables selected by CART and CHAID, this study compares the eight models, and their prediction accuracy from high to low is CHAID–C5.0 (95.65%), CART–C5.0 (92.77%), CART–XGB (89.44%), CHAID–XGB (88.82%), CART–ANN (88.82%), CHAID–ANN (88.67%), CART–SVM (87.86%), and CHAID–SVM (83.54%).

According to Tables 5 and 6, in summary, the order of performance of algorithmic classifiers, the C5.0 is the best (CHAID–C5.0: 95.65%; CART–C5.0: 92.77%), followed by the XGB (CART–XGB: 89.44%; CHAID–XGB: 88.82%), the ANN (CART–ANN: 88.82%; CHAID–ANN: 88.67%), and the SVM (CART–SVM: 87.86%; CHAID–SVM: 83.54%).

This study further evaluates the performance of CHAID–C5.0 and CART–C5.0 by various indicators, such as the confusion matrix (accuracy, precision, recall, and F1-score), the ROC curve, and the AUC value, and has determined that the CHAID–C5.0 model is better than the CART–C5.0 model.

Another noteworthy indicator in judging whether enterprises have going-concern doubts is whether CART or CHAID is used to select the important variables, and return on stockholders' equity, return on total assets, earnings per share, and total liabilities/ stockholders' equity all have high importance values.

## 6. Conclusions

Since the global financial crisis from 2008 to 2009, the financial crisis and bankruptcy of enterprises have emerged endlessly and been prominent, which has caused serious losses to investors, national and regional economies, and even the global economy; thus, the problem of going-concern doubts of enterprises has drawn more attention. In 2001, the Enron financial scandal broke out in the United States, which exposed deficiencies in the accounting and securities regulation of listed companies. The United States Congress quickly passed the Sarbanes–Oxley Act in 2002, which imposes strict requirements and legal liability on CPAs. Therefore, in order to reduce the probability and risk of audit failure for CPAs and auditors, accounting firms pay greater attention to whether enterprises have financial crises or going-concern crises. To appropriately assess and issue audit opinions of going-concern doubts is dependent on the requirements of audit standards and the

professional judgments of CPAs. However, the audit opinions of going-concern doubts involve future uncertainties, and the risks remain very high; thus, accounting firms may consider whether or not to accept appointments if they find out in time that clients have going-concern doubts. Therefore, the construction of a useful and highly-accurate going-concern prediction model is one of the most important and urgent issues in current audit research and auditing practices.

This study first selects important variables with CART and CHAID, and then constructs the going-concern prediction models by three machine-learning algorithms with strong learning and prediction abilities, namely XGB, ANN, SVM, and C5.0. Among the eight models constructed in this study, the prediction accuracy of the CHAID–C5.0 model in the test group is the highest (95.65%), followed by the CART–C5.0 model (92.77%), and the prediction accuracy of all eight models is higher than 83%, while the proportion of GCD samples mispredicted as Non-GCD samples in the total samples in the test group is lower than 9%. In terms of social-science research, this study has good empirical results [1,35]. In addition, whether CART or CHAID are employed to select important variables, return on stockholders' equity, return on total assets, earnings per share, and total liabilities/stockholders' equity have high importance values, and thus are worthy of special attention by CPAs in auditing and making audit decisions.

In the face of mass data and the age of AI, strict going-concern prediction models are constructed in this study by combining hybrid machine learning technologies, namely, CART, CHAID, XGB, ANN, SVM, and C5.0, and a total of eight going-concern prediction models with a prediction accuracy higher than 80% are proposed. This study follows previous research on the application of machine-learning technologies in going concern and constructs eight going-concern prediction models. Among the eight going-concern prediction models constructed in this study, the CHAID–C5.0 model has the highest accuracy of 95.65%, and the lowest accuracy of 83.54% of the CHAID–SVM model, the prediction accuracy rate of all models is higher than 80%. In other words, this study has successfully constructed going-concern prediction models using machine-learning technologies. The results can provide reference for CPAs, auditors, top management, and financial executives to conduct academic research on going concern and auditing, and thus contribute to practice, academic research, and literature extension. To conclude, based on the financial and non-financially important variables that this study screened (debt ratio, return on stockholders' equity, return on total assets, earnings per share, total liabilities/stockholders' equity . . . etc.) and the two-stage models we constructed by hybrid machine-learning technologies, these will help researchers, policymakers, and business owners have a deeper understanding regarding the going-concern prediction.

Regarding the limitation of this study, this study constructs going-concern prediction models with Taiwan's listed and OTC (over-the-counter) companies as the research samples; however, Taiwan's capital market is smaller than that of other global economies [1,16,23,33]. Moreover, in order to apply the going-concern prediction models constructed in this study to countries or regions other than Taiwan, it is necessary to adjust the variables to measure going concern according to the status of local enterprises, financial market conditions, corporate regulations, capital market regulations, and audit laws [16,23]. It is further explained that due to the influence of factors such as economy, finance, capital market, bank financing, and other factors in different countries or regions, the important variables used in this study, such as the "debt ratio" and "audited by BIG4 or not" should be carefully considered whether to use or not. In addition, the results of this study are not applicable to sudden executive frauds, such as the tunneling of companies, or the impact of environmental factors, such as the global financial tsunami or the COVID-19 global crisis [23,36].

**Author Contributions:** Conceptualization, D.-J.C.; methodology, D.-J.C.; software, D.-J.C. and Z.-D.S.; validation, D.-J.C.; formal analysis, D.-J.C.; investigation, D.-J.C. and Z.-D.S.; data curation, D.-J.C. and Z.-D.S.; writing—original draft preparation, D.-J.C. and Z.-D.S.; writing—review and editing, D.-J.C.; visualization, D.-J.C. All authors have read and agreed to the published version of the manuscript.

**Funding:** The research expenditure, and the editing and polishing charges for the English version of this manuscript have been supported by Ministry of Science and Technology, Taiwan, under grant no. MOST 109-2410-H-034-035.

**Institutional Review Board Statement:** Not applicable.

**Informed Consent Statement:** Not applicable.

**Data Availability Statement:** Data can be provided upon request from the corresponding author.

**Acknowledgments:** The authors would like to express their gratitude toward Ministry of Science and Technology, Taiwan, for the subsidy (MOST 109-2410-H-034-035) on this research. They hereby declare that based on academic ethics, they have specified the origin of their data and the sources of the manuscript content and have cited and quoted accordingly. This research is based on the modified version of (including the majority of) the final report which has been subsidized by Ministry of Science and Technology, Taiwan, under grant no. MOST 109-2410-H-034-035 and is elaborated by adding some empirical research and the subsequent empirical results.

**Conflicts of Interest:** The authors declare no conflict of interest.

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
