# Peer review of "Using Hybrid Artificial Intelligence and Machine Learning Technologies for Sustainability in Going-Concern Prediction"

_sustainability, doi:10.3390/su14031810_

Round 1
Reviewer 1 Report
The purpose of this study is to propose suitable methods to construct going concern opinion decision models, in order to detect the signs of bankruptcy in advance and reduce losses to investors and auditors. This study integrates AI and machine learning technologies, including the selection of important variables by two decision trees: the chi-squared automatic interaction detector (CHAID) and classification and regression trees (CART); then, the classification models are respectively constructed by machine learning technologies. The subject addressed is in the scope of Sustainability. However, the new contribution of the present study is not clear. Therefore, this study in the current version needs major revisions. Other problems from my perspective are as follows:
Major revision:
- What is the new contribution of the present study? All of the used machine learning are existing methods. Why do you think that other studies are inadequate? Is your results' accuracy better than before studies, or do you present a new methodology? Please state about them in more detail.
- What is the selected parameter of employed algorithms? How do you determine them? I suggested the parameters of algorithms determined by sensitivity analysis.
- There is a lake of discussing reasons for the superiority of CHAID-C5.0 and CART-XGB. All of the employed algorithms in this study have essential parameters that can impact final results. So that sensitivity analysis of them may change results. Therefore wthout sensitivity analysis can not say that witch algorithm is better.
- Please add the statistical characteristics of inputs and target data to the material and methods.
- Please add kernel function equation in the "Support Vector Machine (SVM)" section.
- The evaluation criterion equation should be mentioned in the material and methods section.
- The results section is too short. Please add more results of modeling. You can add results of histogram errors and scatter plots.
- Why do some input variables have more significance than other input variables. Please explain them in the discussion section.
- How you negotiate the uncertainty of machine learning and its parameters. Please add the results of the uncertainty analysis.
And some minor revisions:
- Please add graphical abstract to the manuscript.
- Please add highlights to the manuscript.
- Please add a flowchart of the CHAID structure to the manuscript.
- The quality of the figures and tables is low.
- The manuscript could be substantially improved by relying and citing more on recent literatures (2021 papers).
Author Response
We greatly appreciate the reviewer to provide the valuable comments, which are very helpful to improve the quality of our paper. The improvements are described as follows:
Point 1: What is the new contribution of the present study? All of the used machine learning are existing methods. Why do you think that other studies are inadequate? Is your results' accuracy better than before studies, or do you present a new methodology? Please state about them in more detail.
Response 1: Thank you for your comments and suggestions.
We have improved our paper in sections “1. Introduction”, “3. Materials and Methods”, “5. Discussion”, and “6. Conclusions”. Please see “1. Introduction”, “3. Materials and Methods”, “5. Discussion”, and “6. Conclusions” highlighted in red.
Point 2: What is the selected parameter of employed algorithms? How do you determine them? I suggested the parameters of algorithms determined by sensitivity analysis.
Response 2: Thank you for your comments and suggestions.
We have explained the selected parameter of employed algorithms in our paper in section “4.2. Model Construction Results”. Please see “4.2. Model Construction Results” highlighted in red.
After referring to several representative papers of machine learning (Jan,2021; 2021; 2021; Chi and Chu,2021), we have improved our paper by adding various performance indicators instead of sensitivity analysis, such as confusion matrix (accuracy, precision, recall, and F1-score), ROC curve and AUC value to evaluate the performance of the best two models in sections “4.3. Additional Comparison of CHAID-C5.0 and CART-C5.0”, “5. Discussion”, and “6. Conclusions”. Please see “4.3. Additional Comparison of CHAID-C5.0 and CART-C5.0”, “5. Discussion”, and “6. Conclusions” highlighted in red.
- Jan, C.L. Using deep learning algorithms for CPAs’ going concern prediction. Information 2021, 12, 73.
- Chi D.J.; Chu, C.C. Artificial intelligence in corporate sustainability: Using LSTM and GRU for going concern prediction. Sustainability. 2021, 13, 11631.
- Jan, C.L. Financial information asymmetry: Using deep learning algorithms to predict financial distress. Symmetry 2021, 13, 443.
- Jan, C.L. Detection of financial statement fraud using deep learning for sustainable development of capital markets under information asymmetry. Sustainability 2021, 13, 9879.
Point 3: There is a lake of discussing reasons for the superiority of CHAID-C5.0 and CART-XGB.
Response 3: Thank you for your comments and suggestions.
We have corrected the best two models as CHAID-C5.0 and CART-C5.0 (not CART-XGB) and to improve our paper by adding section “4.3. Additional Comparison of CHAID-C5.0 and CART-C5.0”. Please see “4.3. Additional Comparison of CHAID-C5.0 and CART-C5.0” highlighted in red.
Point 4: All of the employed algorithms in this study have essential parameters that can impact final results. So that sensitivity analysis of them may change results. Therefore wthout sensitivity analysis can not say that witch algorithm is better.
Response 4: Thank you for your comments and suggestions.
We have improved our paper our paper in sections “4.2. Model Construction Results” and “4.3. Additional Comparison of CHAID-C5.0 and CART-C5.0”. Please see “4.2. Model Construction Results” and “4.3. Additional Comparison of CHAID-C5.0 and CART-C5.0” highlighted in red.
After referring to several representative papers of machine learning (Jan,2021; 2021; 2021; Chi and Chu,2021), we have improved our paper by adding various performance indicators instead of sensitivity analysis, such as confusion matrix (accuracy, precision, recall, and F1-score), ROC curve and AUC value to evaluate the performance of the best two models in sections “4.3. Additional Comparison of CHAID-C5.0 and CART-C5.0”, “5. Discussion”, and “6. Conclusions”. Please see “4.3. Additional Comparison of CHAID-C5.0 and CART-C5.0”, “5. Discussion”, and “6. Conclusions” highlighted in red.
- Jan, C.L. Using deep learning algorithms for CPAs’ going concern prediction. Information 2021, 12, 73.
- Chi D.J.; Chu, C.C. Artificial intelligence in corporate sustainability: Using LSTM and GRU for going concern prediction. Sustainability. 2021, 13, 11631.
- Jan, C.L. Financial information asymmetry: Using deep learning algorithms to predict financial distress. Symmetry 2021, 13, 443.
- Jan, C.L. Detection of financial statement fraud using deep learning for sustainable development of capital markets under information asymmetry. Sustainability 2021, 13, 9879.
Point 5: Please add the statistical characteristics of inputs and target data to the material and methods.
Response 5: Thank you for your comments and suggestions.
We have improved our paper by adding section “3.8.2. Variable Characteristics and Definitions”. Please see “3.8.2. Variable Characteristics and Definitions” highlighted in red.
Point 6: Please add kernel function equation in the "Support Vector Machine (SVM)" section.
Response 6: Thank you for your comments and suggestions.
We have added kernel function equation in section “3.5. Support Vector Machine (SVM)”. Please see “3.5. Support Vector Machine (SVM)” highlighted in red.
Point 7: The evaluation criterion equation should be mentioned in the material and methods section.
Response 7: Thank you for your comments and suggestions.
We have added the evaluation criterion equations in new added section “3.7 Models’ Performance Evaluation Methods”. Please see “3.7 Models’ Performance Evaluation Methods” highlighted in red.
Point 8: The results section is too short. Please add more results of modeling. You can add results of histogram errors and scatter plots.
Response 8: Thank you for your comments and suggestions.
After referring to several representative papers of machine learning (Jan,2021; 2021; 2021; Chi and Chu,2021), we have improved our paper by adding various performance indicators such as confusion matrix (accuracy, precision, recall, and F1-score), ROC curve and AUC value to evaluate the performance of the best two models in sections “4.3. Additional Comparison of CHAID-C5.0 and CART-C5.0”, “5. Discussion”, and “6. Conclusions”. Please see “4.3. Additional Comparison of CHAID-C5.0 and CART-C5.0”, “5. Discussion”, and “6. Conclusions” highlighted in red.
- Jan, C.L. Using deep learning algorithms for CPAs’ going concern prediction. Information 2021, 12, 73.
- Chi D.J.; Chu, C.C. Artificial intelligence in corporate sustainability: Using LSTM and GRU for going concern prediction. Sustainability. 2021, 13, 11631.
- Jan, C.L. Financial information asymmetry: Using deep learning algorithms to predict financial distress. Symmetry 2021, 13, 443.
- Jan, C.L. Detection of financial statement fraud using deep learning for sustainable development of capital markets under information asymmetry. Sustainability 2021, 13, 9879.
Point 9: Why do some input variables have more significance than other input variables. Please explain them in the discussion section.
Response 9: Thank you for your comments and suggestions.
We have explained them in sections “3. Materials and Methods”, “4. Results”, “5. Discussion”, and “6. Conclusions”. Please see “3. Materials and Methods”, “4. Results”, “5. Discussion”, and “6. Conclusions” highlighted in red.
Point 10: How you negotiate the uncertainty of machine learning and its parameters. Please add the results of the uncertainty analysis.
Response 10: Thank you for your comments and suggestions.
We use the pre-set parameters of the software and adjust some parameters to obtain the best performance of the models. Therefore, there is no need to conduct an uncertainty analysis for this study. Parameters are listed in section “4.2. Model Construction Results”. Please see “4.2. Model Construction Results” highlighted in red.
Point 11: Please add graphical abstract to the manuscript.
Response 11: Thank you for your comments and suggestions.
We would like to ask your forgiveness! After overviewing the published articles on Sustainability, we do not add graphical abstract for simplicity and clarity.
Point 12: Please add highlights to the manuscript.
Response 12: Thank you for your comments and suggestions.
We have added highlights of this study in the last two paragraphs of section “1. Introduction”. Please see the last two paragraphs of section “1. Introduction” highlighted in red.
Point 13: Please add a flowchart of the CHAID structure to the manuscript.
Response 13: Thank you for your comments and suggestions.
We have added a flowchart of the CHAID structure in section “4.2.2. CHAID Models” (Figure 2. Decision path and structure of the CHAID-C5.0 model). Please see “4.2.2. CHAID Models” (Figure 2. Decision path and structure of the CHAID-C5.0 model) highlighted in red.
Point 14: The quality of the figures and tables is low.
Response 14: Thank you for your comments and suggestions.
We have improved the quality of all the figures and tables in our paper.
Point 15: The manuscript could be substantially improved by relying and citing more on recent literatures (2021 papers).
Response 15: Thank you for your comments and suggestions.
We have added 8 papers (2017-2022) in our paper. Please see the sections “1. Introduction”, “2. Literature Review”, “3. Materials and Methods”, and “References” highlighted in red.
- Agostini, M. The role of going concern evaluation in both prediction and explanation of corporate financial distress: Concluding remarks and future trends. Corporate Financial Distress, Palgrave Macmillan: London, United Kingdom, 2018, 119–126.
- Hategan, C.D.; Imbrescu, C.M. Auditor's uncertainty about going concern predictor of insolvency risk. Ovidius Univ. Ann. Econ. Sci. Ser. 2018, 18(2), 605–610.
- Barboza, F.; Kimura, H.; Altman, E. Machine learning models and bankruptcy prediction. Expert Syst. Appl. 2017, 83, 405-417.
- PaweÅ‚ek, B. Extreme gradient boosting method in the prediction of company bankruptcy. Stats. Trans. New Ser. (SiTns), 2019, 20(2), 155–171.
- Chen, S.; Shen, Z.D. Financial distress prediction using hybrid machine learning techniques. Asian J. Econ. Bus. Account. 2020, 1–12.
- Faysal, J.A.; Mostafa, S.T.; Tamanna, J.S.; Mumenin, K.M.; Arifin, M.M.; Awal, M.A.; Shome A.; Mostafa, S.S. XGB-RF: A hybrid machine learning approach for IoT intrusion detection. Telecom, 2022, 3, 52–69. https://doi.org/10.3390/ telecom3010003
- Chen, S.; Jhuang, S. Financial distress prediction using data mining techniques. ICIC-ELB. 2018, 9(2), 131–136.
- Jan, C.L. Financial information asymmetry: Using deep learning algorithms to predict financial distress. Symmetry 2021, 13, 443.
Sincerely thank you again for your valuable comments!

Reviewer 2 Report
The paper tests the suitable methods to construct going concern opinion decision models. The topic is interesting, but the article needs some improvement.
In the Introduction section, the paper's purpose should be better explained. Also, this section needs to be added to the sentence with the paper structure.
Going concern is a hotly debated topic; the literature needs to be updated with recent publications from the last 5 years; please see a list below or search other relevant papers.
Agostini, M. (2018). The Role of Going Concern Evaluation in Both Prediction and Explanation of Corporate Financial Distress: Concluding Remarks and Future Trends. Corporate Financial Distress, 119-126.
Barboza, F., Kimura, H., & Altman, E. (2017). Machine learning models and bankruptcy prediction. Expert Systems with Applications, 83, 405-417.
Chen, S., & Shen, Z. D. (2020). Financial Distress Prediction Using Hybrid Machine Learning Techniques. Asian Journal of Economics, Business and Accounting, 1-12.
Hategan C.D., Imbrescu C.M. (2018). Auditor's Uncertainty About Going Concern Predictor of Insolvency Risk. Ovidius University Annals, Economic Sciences Series, 18(2), 605-610.
Pawełek, B. (2019). Extreme Gradient Boosting Method in the Prediction of Company Bankruptcy. Statistics in Transition. New Series, 20(2), 155-171.
The author(s) need to cite other studies to support the selection of the qualitative and quantitative indicators included in the model.
The research implications are very general, and it is not clear how the study will help researchers, policymakers, and business owners have a deeper understanding regarding the going concern prediction.
The results must be discussed according to previous research. Also, it can be discussed how the results can be generalized to companies from other countries based on the literature if there are gaps between companies from different countries.
Minor aspects
Please provide full name for all abbreviations, f.e. OTC
Author Response
Response to Reviewer 2 Comments
We greatly appreciate the reviewer to provide the valuable comments, which are very helpful to improve the quality of our paper. The improvements are described as follows:
Point 1: In the Introduction section, the paper's purpose should be better explained. Also, this section needs to be added to the sentence with the paper structure.
Response 1: Thank you for your comments and suggestions.
We have improved our paper in section “1. Introduction”. Please see the last two paragraphs of the section “1. Introduction” highlighted in red.
Point 2: Going concern is a hotly debated topic; the literature needs to be updated with recent publications from the last 5 years; please see a list below or search other relevant papers.
Agostini, M. (2018). The Role of Going Concern Evaluation in Both Prediction and Explanation of Corporate Financial Distress: Concluding Remarks and Future Trends. Corporate Financial Distress, 119-126.
Barboza, F., Kimura, H., & Altman, E. (2017). Machine learning models and bankruptcy prediction. Expert Systems with Applications, 83, 405-417.
Chen, S., & Shen, Z. D. (2020). Financial Distress Prediction Using Hybrid Machine Learning Techniques. Asian Journal of Economics, Business and Accounting, 1-12.
Hategan C.D., Imbrescu C.M. (2018). Auditor's Uncertainty About Going Concern Predictor of Insolvency Risk. Ovidius University Annals, Economic Sciences Series, 18(2), 605-610.
Pawełek, B. (2019). Extreme Gradient Boosting Method in the Prediction of Company Bankruptcy. Statistics in Transition. New Series, 20(2), 155-171.
Response 2: Thank you for your comments and suggestions.
We have cited these papers in in sections “2. Literature Review” and “3. Materials and Methods”. Please see “2. Literature Review”, “3. Materials and Methods”, and “References” highlighted in red.
- Agostini, M. The role of going concern evaluation in both prediction and explanation of corporate financial distress: Concluding remarks and future trends. Corporate Financial Distress, Palgrave Macmillan: London, United Kingdom, 2018, 119–126.
- Hategan, C.D.; Imbrescu, C.M. Auditor's uncertainty about going concern predictor of insolvency risk. Ovidius Univ. Ann. Econ. Sci. Ser. 2018, 18(2), 605–610.
- Barboza, F.; Kimura, H.; Altman, E. Machine learning models and bankruptcy prediction. Expert Syst. Appl. 2017, 83, 405-417.
- PaweÅ‚ek, B. Extreme gradient boosting method in the prediction of company bankruptcy. Stats. Trans. New Ser. (SiTns), 2019, 20(2), 155–171.
- Chen, S.; Shen, Z.D. Financial distress prediction using hybrid machine learning techniques. Asian J. Econ. Bus. Account. 2020, 1–12.
- Faysal, J.A.; Mostafa, S.T.; Tamanna, J.S.; Mumenin, K.M.; Arifin, M.M.; Awal, M.A.; Shome A.; Mostafa, S.S. XGB-RF: A hybrid machine learning approach for IoT intrusion detection. Telecom, 2022, 3, 52–69. https://doi.org/10.3390/ telecom3010003
- Chen, S.; Jhuang, S. Financial distress prediction using data mining techniques. ICIC-ELB. 2018, 9(2), 131–136.
- Jan, C.L. Financial information asymmetry: Using deep learning algorithms to predict financial distress. Symmetry 2021, 13, 443.
Point 3: The author(s) need to cite other studies to support the selection of the qualitative and quantitative indicators included in the model.
Response 3: Thank you for your comments and suggestions.
We have cited other studies to support the selection of the qualitative and quantitative indicators included in the model. Please see the right column “Sources” of Table 2 highlighted in red.
Point 4: The research implications are very general, and it is not clear how the study will help researchers, policymakers, and business owners have a deeper understanding regarding the going concern prediction.
Response 4: Thank you for your comments and suggestions.
We have improved our paper in sections “1. Introduction” and “6. Conclusions”. Please see sections “1. Introduction” and “6. Conclusions” highlighted in red.
Point 5: The results must be discussed according to previous research. Also, it can be discussed how the results can be generalized to companies from other countries based on the literature if there are gaps between companies from different countries.
Response 5: Thank you for your comments and suggestions.
We have improved our paper in sections “5. Discussion” and “6. Conclusions”. Please see sections “5. Discussion” and “6. Conclusions” highlighted in red.
Point 6: Please provide full name for all abbreviations, f.e. OTC
Response 6: Thank you for your comments and suggestions.
We have provided full name for all abbreviations, f.e. OTC (over-the-counter), CPAs (certified public accountants), AI (artificial intelligence), key audit matters (KAMs).
Sincerely thank you again for your valuable comments!

Reviewer 3 Report
The paper develops an interesting topic but it must include more actual references. Most of the bibliographic references in the paper are older than 20 years.
Moreover, authors should rely upon the theoretical background to explain the significance of their results.
Author Response
Response to Reviewer 3 Comments
We greatly appreciate the reviewer to provide the valuable comments, which are very helpful to improve the quality of our paper. The improvements are described as follows:
Point 1: The paper develops an interesting topic but it must include more actual references. Most of the bibliographic references in the paper are older than 20 years.
Response 1: Thank you for your comments and suggestions.
We have cited some papers (2017-2022) in sections “2. Literature Review” and “3. Materials and Methods”. Please see “2. Literature Review”, “3. Materials and Methods”, and “References” highlighted in red.
- Agostini, M. The role of going concern evaluation in both prediction and explanation of corporate financial distress: Concluding remarks and future trends. Corporate Financial Distress, Palgrave Macmillan: London, United Kingdom, 2018, 119–126.
- Hategan, C.D.; Imbrescu, C.M. Auditor's uncertainty about going concern predictor of insolvency risk. Ovidius Univ. Ann. Econ. Sci. Ser. 2018, 18(2), 605–610.
- Barboza, F.; Kimura, H.; Altman, E. Machine learning models and bankruptcy prediction. Expert Syst. Appl. 2017, 83, 405-417.
- PaweÅ‚ek, B. Extreme gradient boosting method in the prediction of company bankruptcy. Stats. Trans. New Ser. (SiTns), 2019, 20(2), 155–171.
- Chen, S.; Shen, Z.D. Financial distress prediction using hybrid machine learning techniques. Asian J. Econ. Bus. Account. 2020, 1–12.
- Faysal, J.A.; Mostafa, S.T.; Tamanna, J.S.; Mumenin, K.M.; Arifin, M.M.; Awal, M.A.; Shome A.; Mostafa, S.S. XGB-RF: A hybrid machine learning approach for IoT intrusion detection. Telecom, 2022, 3, 52–69. https://doi.org/10.3390/ telecom3010003
- Chen, S.; Jhuang, S. Financial distress prediction using data mining techniques. ICIC-ELB. 2018, 9(2), 131–136.
- Jan, C.L. Financial information asymmetry: Using deep learning algorithms to predict financial distress. Symmetry 2021, 13, 443.
Point 2: Moreover, authors should rely upon the theoretical background to explain the significance of their results.
Response 2: Thank you for your comments and suggestions.
We have improved our paper in sections “5. Discussion” and “6. Conclusions”. Please see “5. Discussion” and “6. Conclusions” highlighted in red.
Sincerely thank you again for your valuable comments!

Round 2
Reviewer 1 Report
Accept
Reviewer 2 Report
The paper is improved, all recommendations have been taken into account.